# A Cellulose Electrolysis Cell with Metal-Free Carbon Electrodes

**Yaorong Li [1], Masahiro Nagao [1], Kazuyo Kobayashi [1], Yongcheng Jin [2] and Takashi Hibino [1,***

[1]  Graduate School of Environmental Studies, Nagoya University, Nagoya 464-8601, Japan;
    li.yaorong@e.mbox.nagoya-u.ac.jp (Y.L.); nagao@urban.env.nagoya-u.ac.jp (M.N.);
    kkoba@urban.env.nagoya-u.ac.jp (K.K.)
[2]  School of Materials Science and Engineering, Ocean University of China, Qingdao 266100, China;
    jinyongcheng@ouc.edu.cn
*   Correspondence: hibino@urban.env.nagoya-u.ac.jp

**Abstract:** Biomass raw materials, including agricultural residues, collected weeds, and wood chips, are important feedstocks for hydrogen production. Numerous attempts have been made to electrolyze biomass directly or indirectly to hydrogen because these processes allow for the production of hydrogen with less power consumption than water electrolysis. However, expensive metal-based electrocatalysts are needed, especially for the cathode reaction, in the electrolysis cells. Results from the present study demonstrate the production of hydrogen directly from cellulose, using an optimal mesoporous carbon as the cathode in addition to a partially oxygenated carbon anode at a temperature of 150 °C, with an electrolysis onset voltage of ca. 0.2 V, a current density of 0.29 A cm$^{-2}$ at an electrolysis voltage of 1 V, and a current efficiency of approximately 100% for hydrogen production. These characteristics were comparable to those recorded when using a Pt/C anode and cathode under the same conditions. The sp$^2$ planes of the carbon allowed π electrons to be donated to protons at the cathode. In addition, the mesoporous structure provided a sufficient amount of sp$^2$ planes on the surface of the cathode.

**Keywords:** cellulose; electrolysis; hydrogen production; metal-free electrodes

## 1. Introduction

Hydrogen is a cleaner energy source and more reliable carrier than fossil fuels [1–3]. A major challenge in the utilization of hydrogen as a fuel is the lack of sustainable production methods [4,5]. Water electrolysis ensures low-carbon-footprint hydrogen production using renewable energy resources, but has limitations of high cost; the electricity energy required is approximately 5 kWh (Nm$^3$)$^{-1}$ [6]. Furthermore, the use of precious metals as electrocatalysts is needed for the anode and cathode reactions [7]. Operating an electrolysis cell at elevated temperatures provides some advantages over water electrolysis at room temperature [8,9]. The onset cell voltage decreased as temperature increases (1.16 V at 130 °C), due to the decrease in the enthalpy required for water splitting [10,11]. In addition, at elevated temperatures, various biomass derivatives, including alcohols, ethylene glycol, glycerol, and lignin, can also be used as hydrogen resources, which enables hydrogen production at lower onset cell voltages (0.4–0.5 V at 80–150 °C), depending on the fuel species or operation temperature [12–14].

Electrolysis at elevated temperatures is also advantageous for the kinetic characteristics of the used electrodes. A recent report indicated a lignocellulosic biomass electrolysis cell using partially oxygenated carbon as the anode at temperatures above 125 °C [15,16]. However, Pt was still necessary for the hydrogen evolution reaction (HER) at the cathode in this cell. Although carbides, sulfides, and phosphides of transition metals (e.g., Mo [17,18], W [19], and Co [20,21]) have received attention as

important HER catalysts, their material costs are not much less than that of Pt. The present study describes an inexpensive mesoporous carbon that exhibited HER activity comparable to that obtained for a Pt/C cathode. Consequently, efficient hydrogen production was achieved using cellulose as a model biomass at temperatures between 100 and 150 °C in an electrolysis cell with a metal-free anode and cathode.

## 2. Results and Discussion

The partially oxygenated carbon anode used in this study was synthesized according to a procedure reported previously [15,16]. Briefly, 1.0 g of Ketjen Black (KB) was stirred in 24% $HNO_3$ diluted to 50 mL at room temperature for 74 h. After filtering and washing, the oxygenated KB was heated at 600 °C for 4 h under an Ar flow to eliminate carboxyl groups preferentially (as opposed to carbonyl groups having higher thermal stability) from the carbon surface. First, the cellulose electrolysis process was assessed using a partially oxygenated KB anode and Pt/C cathode. The current–voltage (I–V) curves were measured with and without the cellulose fuel at a temperature of 150 °C (Figure 1a). The electrolysis onset voltage of the cell was ca. 1.2 V for the blank, but was ca. 0.3 V for the cellulose. Accordingly, cellulose rather than water and $H_3PO_4$ is hydrogenated at a cell voltages less than 1 V. Another important result is that the I–V characteristics of the cell using the partially oxygenated KB anode were equal to those of the cell using a control Pt/C anode, indicating that the carbonyl groups are important active sites for the anode reaction. Analysis of the outlet gases from the anode and cathode demonstrated that the formation rates of hydrogen and $CO_2$ at the cathode and anode, respectively, were nearly proportional to the current density (Figure 1b). Theoretical formation rates for hydrogen and $CO_2$ were calculated based on the following the following reaction equations [22].

$$\text{Anode: } C_6H_{12}O_6 + 6H_2O \rightarrow 6CO_2 + 24H^+ + 24e^- \tag{1}$$

$$\text{Cathode: } 24H^+ + 24e^- \rightarrow 12H_2 \tag{2}$$

(Cellulose was hydrolyzed to monosaccharide derivatives without excessive oxidation by adding $H_3PO_4$ to the anode [23]). Estimated current efficiencies were approximately 100% and 90% for hydrogen and $CO_2$ formation, respectively. Therefore, the oxidation reaction of the decomposition products of cellulose contributed to the low onset cell voltage observed for cellulose. The I–V characteristics were dependent on the operational temperature (Figure 1c): the electrolysis onset voltage increased as temperature decreased; the current density at each voltage decreased with temperature. The impedance characteristics were also dependent on operational temperature (Figure 1d; data were recorded at an anodic bias voltage of 0.4 V). The axis intersection of the impedance plot at high frequency, attributed to the ohmic resistance of the electrolyte, shifted to higher impedance values as the temperature decreased. The impedance plot at medium and low frequencies was affected by temperature in a more complex manner. Semicircular arcs observed at 125 and 150 °C were related to the charge and mass transfer resistances of the electrodes. By contrast, impedance plots recorded at 75 and 100 °C were nearly linear, indicating a capacitive behavior [24,25]. These results demonstrate that the electrode reactions did not begin at an anodic bias voltage of 0.4 V at temperatures of 100 °C or lower, which corresponds to the electrolysis onset voltage observed in the I–V characteristics.

Carbon blacks were initially investigated as cathode candidates, owing to their low price and good electrical conductivity. π electron radicals of the carbon basal planes are also susceptible to electrophilic reactions [26,27], which include HER. Acetylene black (AB), Vulcan, and KB were tested as possible cathode catalysts at a temperature of 150 °C. The I–V characteristics increased in the order of KB > Vulcan > AB (Figure S1a), which agrees especially well with the order of pore volume and average pore diameter (Table S1 [28,29]) that influences the gas removability and ionomer wettability of the electrode, as reflected by the impedance spectra of the electrolysis cells using these cathodes (Figure S1b).

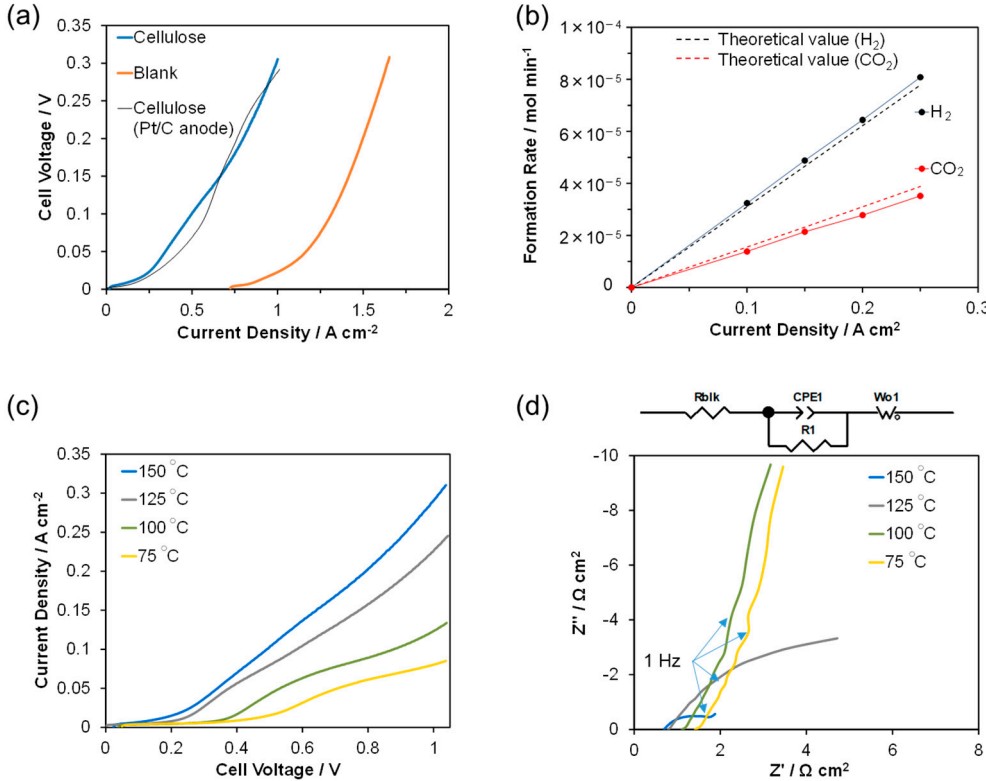

**Figure 1.** Characteristics for cellulose electrolysis of the cell using a partially oxygenated Ketjen Black (KB) anode and Pt/C cathode: (**a**) I–V curves at 150 °C with and without cellulose. Data for an electrolysis cell using a Pt/C anode and cathode are included for comparison. (**b**) Formation rates of hydrogen at the cathode and $CO_2$ at the anode. Theoretical values are shown by the dotted lines. (**c**) I–V curves and (**d**) Nyquist impedance plots between 75 and 150 °C.

To examine carbon species with different pore structures, commercially available mesoporous carbon materials (MH, MJ10, 30, and 150) were used in place of carbon blacks. The porosity of the four samples was adjusted by controlling the amount of MgO used as the template [30,31]. The adsorption/desorption isotherms obtained for the samples (including KB) presented H3-type hysteresis loops [32]; low-pressure closure of the loops depended significantly on the carbon species (Figure 2a). Thus, the four samples showed characteristic pore size distributions, all of which were different from that observed for KB (Figure 2b; meso- and macropore properties are summarized in Table 1). The MJ30 and 150 possessed additional macropores of 100 nm or larger, which were visible on Scanning electron microscopy (SEM) micrographs (Figure S2). More importantly, MH and MJ10 exhibited much narrower pore size distributions than KB. Transmission electron microscopy (TEM) revealed that both KB and MJ10 had pores with diameters greater than 10 nm, but that the mesopores of MJ10 were more ordered in their structure than those of KB (Figure 2c,d).

The I–V curves recorded for the four cathode samples indicated that the electrolysis onset voltage was ca. 0.2 V, which is slightly lower than the value observed for the KB cathode (Figure 3a). Use of the MJ10 cathode resulted in the greatest current densities throughout the voltage range. This high performance was attributable to the low ohmic, charge, and mass transfer resistances of the cathode (Figure S3 and Table S2). However, the charge and mass transfer resistances more significantly affected the I–V characteristics compared to the ohmic resistance; observed ranges of the ohmic, charge, and mass transfer resistances were 1.182–1.464 $\Omega$ cm$^2$, 0.229–10.33 $\Omega$ cm$^2$, and 28.12–60.99 $\Omega$ cm$^2$, respectively. Considering that the specific surface area of MJ10 was smaller than those obtained for MH and KB (Table 1), optimal pore volume and size appear to exist for the HER kinetics. Similar results were obtained for the oxygen reduction reaction (ORR) [33–35]; however, most of these were

related to the removal of water with larger molecular or cluster size than the hydrogen molecules from the cathode. By contrast, MJ10 had the smallest degree of structural deformation among the carbon samples tested, based on the intensity ratio of the D-band to the G-band determined by Raman spectroscopy measurements (Figure S4). This indicates a high proportion of graphitic basal planes in the sample. Therefore, another possible explanation for the high HER activity of MJ10 is its ability to donate unpaired electrons to protons at the cathode, as will be discussed later. One important feature of the MJ10 cathode was the large effect of sample amount on the I–V characteristics (Figure 3b). The current densities obtained for 20 mg cm$^{-2}$ of MJ10 at all the tested voltages were comparable to the values found for the Pt/C cathode under the same conditions. The quantities of hydrogen formed at the cathode were then measured at various current densities. Constant quantities of hydrogen were produced continuously at all the tested current densities (Figure 3c). Notably, the formation rates of hydrogen were similar to the theoretical values in the current—density range from 0.05 to 0.25 A cm$^2$ (Figure 3d). These results demonstrate that hydrogen could be produced efficiently from cellulose without using metal catalysts as both the anode and cathode. In future work, a flow type cell will need to be used for the evaluation of the long-term performance of these electrodes.

To confirm whether the electron structure of carbon can contribute to the onset of the HER, electron spin resonance (ESR) measurements were carried out for MJ10 and KB at room temperature. The two samples gave a similar signal with a *g* value of ca. 2.002, indicating the existence of unpaired electrons (Figure 4). However, there were differences in the intensity and linewidth of these ESR signals: the intensity ratio of MJ10 to KB was 0.14:1; the linewidths were 3.3 mT for MJ10 and 1.6 mT for KB. The signal intensity increases with the content of sp$^3$ hybrid bonds in the sample [36]. The signal linewidth is associated with $\pi$ electron radicals [37]. Therefore, the high HER activity of MJ10 is attributable to the isolated $\pi$ electron radicals rather than the sp$^3$ hybridized dangling bonds. It is also proposed that the mesoporous structure of MJ10 is favorable for gaining a sufficient amount of $\pi$ electron radicals, due to its relatively high specific surface area.

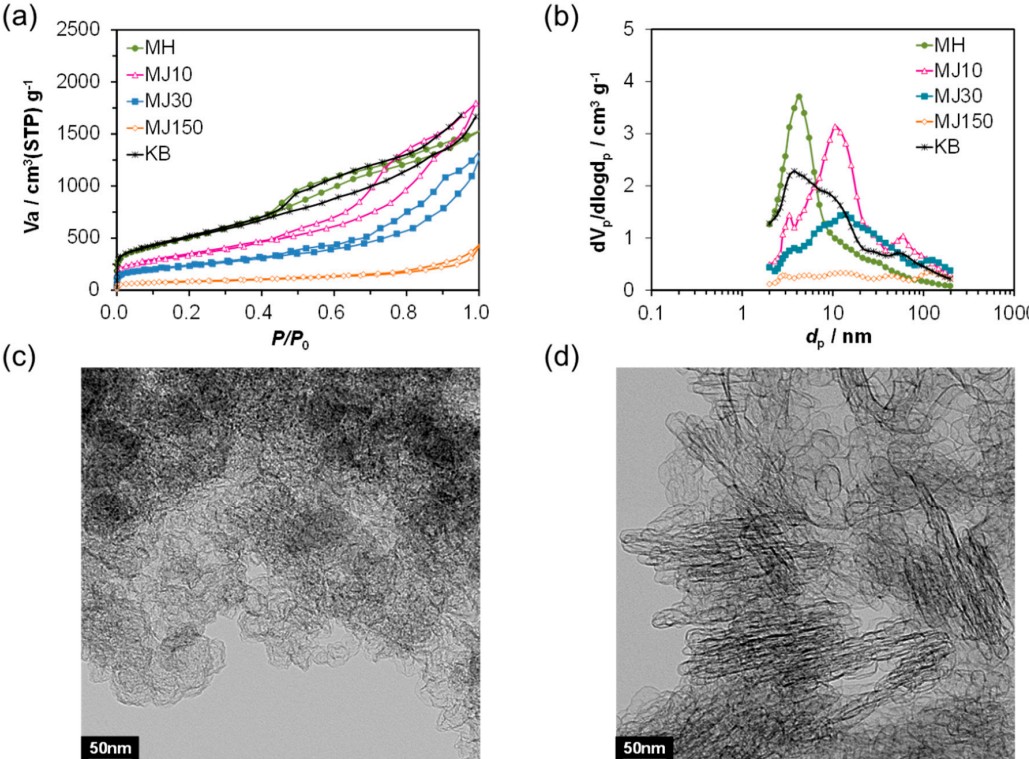

**Figure 2.** Characterization of KB, MH, MJ10, MJ30, and MJ150: (**a**) Nitrogen adsorption/desorption isotherms at −196 °C and (**b**) Barrett–Joyner–Halenda (BJH) pore size distributions. Transmission electron microscopy (TEM) micrographs of (**c**) KB and (**d**) MJ10.

**Table 1.** Textural properties of KB, MH, MJ10, MJ30, and MJ150.

|  | $V_p$ (cm$^3$ g$^{-1}$) (BJH) | $d_p$ (nm) (BJH) | $d_{av}$ (nm) (BJH) | $S_{BET}$ (m$^2$ g$^{-1}$) (BET) |
|---|---|---|---|---|
| **KB** | 2.40 | 3.80 | 5.90 | 1830 |
| **MH** | 2.25 | 4.20 | 4.80 | 1875 |
| **MJ10** | 2.71 | 10.6 | 8.60 | 1223 |
| **MJ30** | 1.76 | 13.9 | 9.30 | 830 |
| **MJ150** | 0.53 | - | 10.0 | 282 |

$V_p$ = pore volume determined using the BJH method; $d_p$ = BJH pore diameter; $d_{av}$ = BJH pore average diameter; $S_{BET}$ = specific surface area determined using the Brunauer–Emmett–Teller (BET) method.

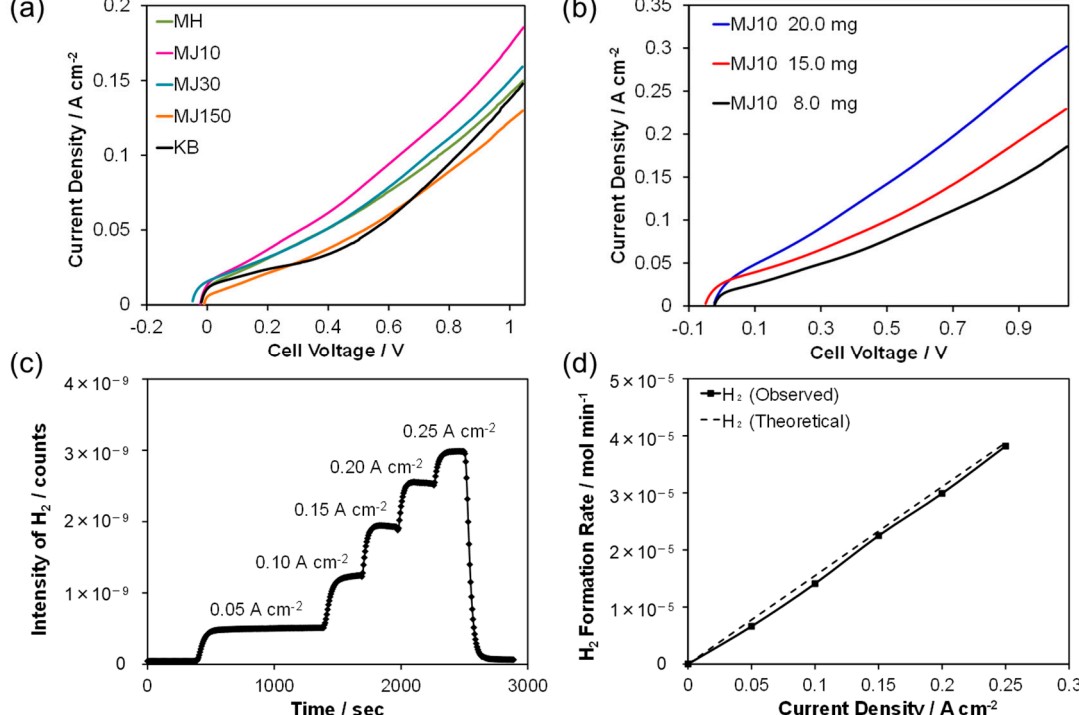

**Figure 3.** Characteristics for cellulose electrolysis of cells using carbon cathodes at 150 °C: I–V curves of (**a**) cells using KB, MH, MJ10, MJ30, and MJ150 cathodes and of (**b**) the cell using the MJ10 cathode. (**c**) Mass spectrometer (MS) spectra for hydrogen and (**d**) hydrogen formation rates at the MJ10 cathode. Theoretical values are shown by the dotted lines.

In conclusion, a partially oxygenated carbon and a structurally optimized carbon could act as the anode and cathode, respectively, in place of Pt/C. In particular, the present study focused on the design of a carbon cathode and determination of its structural and electronic effects. A porous carbon prepared using an MgO-template method had a partially ordered and highly graphitic mesoporous structure, which catalyzed hydrogen evolution on a scale comparable to that found for a Pt/C cathode.

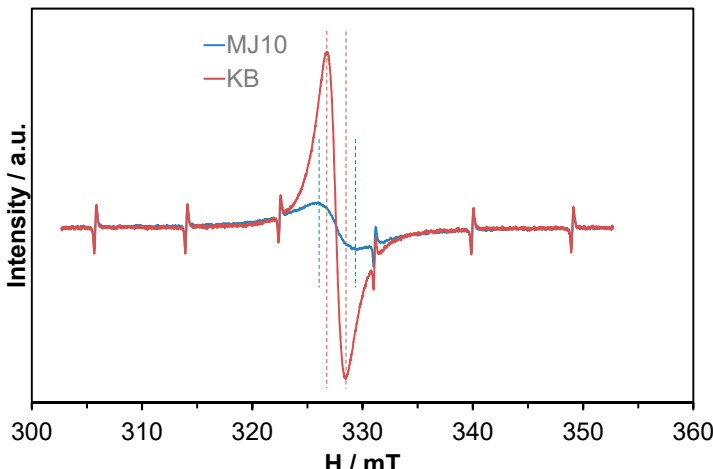

**Figure 4.** ESR signals of KB and MJ10 at room temperature under vacuum.

## 3. Materials and Methods

Vulcan XC-72R, KB EC-600JDK, AB, and MgO-templated porous carbons (MJ10, MJ30, MJ150, and MH) were purchased from Cabot (Billerica, MA, USA), Akzo Nobel (Amsterdam, Netherlands), Denka (Tokyo, Japan), and Toyo Tanso (Osaka, Japan), respectively. The carbon samples were mixed with 85% $H_3PO_4$ (Wako Chemicals, Osaka, Japan) in a mixer (Sankyo Seiki, Thinky AR-100, Kyoto, Japan) for 15 min. The mixture was printed on the surface of a gas diffusion electrode (Toray, TGP-H-090, Tokyo, Japan). All carbon loadings were adjusted to 8 mg cm$^{-2}$, unless otherwise stated. A Pt/C (Electrochem, Woburn, MA, USA, Pt loading: 2 mg cm$^{-2}$) was employed as a control anode and cathode. A proton-conducting $Sn_{0.9}In_{0.1}P_2O_7$-polytetrafluoroethylene (PTFE, Mitsui-Du Pont, Tokyo, Japan) membrane (thickness: 200 μm) was selected as the electrolyte, as reported previously [38,39]. Cellulose (Wako Chemicals, Osaka, Japan) was used as the model biomass without further purification.

Nitrogen isotherms at −196 °C were obtained using a Nihon Bell Belsorp-28SA instrument (Osaka, Japan) after degassing at 200 °C. SEM and TEM micrographs were taken using Keyence VE-8800 (Osaka, Japan) and Jeol JEM2100F (Tokyo, Japan) instruments, respectively. Raman spectra were recorded using a Jasco NRS-1000 spectrophotometer (Tokyo, Japan). ESR measurements were performed using a Jeol JES-FE3XG instrument with a $Mn^{2+}$ marker. The samples were evacuated in a quartz tube at room temperature. The outlet gases from the electrodes were analyzed using a Pfeiffer Vacuum ThermoStar mass spectrometer (MS; Asslar, Germany).

Electrolysis cells were fabricated as follows. Cellulose (15 mg) was impregnated with 85% $H_3PO_4$ (ca. 270 mg), and the paste was stored on the surface of the anode in a similar manner to that reported previously [15,16]. The electrolyte membrane was sandwiched between the anode and cathode with effective areas of 2.0 and 0.5 cm$^2$, respectively. The cathode was exposed to Ar at a flow rate of 100 mL min$^{-1}$. Electrolysis of cellulose was conducted by the four-probe method using a Solartron 1287 potentiostat and a Solartron 1260 impedance/gain-phase analyzer (Hampshire, UK). The I–V curves were recorded at a scan rate of 20 mV s$^{-1}$. Impedance spectra were obtained at an anodic bias voltage of 0.4 V in the frequency range from 0.1 to 10$^6$ Hz. An equivalent circuit model was used for the fitting of the impedance data.

**Supplementary Materials:** The following are available online at http://www.mdpi.com/2073-4344/10/1/106/s1, Figure S1: Electrolysis characteristics of cells using KB, AB, and Vulcan cathodes at 150 °C: (a) I–V curves and (b) impedance spectra. The blank test was conducted without the carbon catalyst at the cathode, Figure S2: SEM micrographs of (a) MH, (b) MJ10, (c) MJ30, and (d) MJ150, Figure S3: Impedance spectra of cells using KB, MH, MJ10, MJ30, and MJ150 cathodes at 150 °C. Black dotted lines represent curve-fitting results, Figure S4: Raman spectra of KB, MH, MJ10, MJ30, and MJ150, Table S1: Textural properties of AB, Vulcan, and KB, Table S2: Fitted values for $R_{bulk}$, $R_1$, and W of cells using the MH, MJ10, MJ30, and MJ150 cathodes.

**Author Contributions:** Conceptualization, T.H.; methodology, T.H., M.N., and Y.J.; investigation, Y.L. and K.K.; data analysis, all authors; writing—original draft preparation, Y.L.; writing—review and editing, T.H., M.N., and Y.J.; supervision, T.H.; funding acquisition, M.N. and T.H. All authors have read and agreed to the published version of the manuscript.

**Funding:** This research was funded by a Kakenhi Grant-in-Aid (Nos. 17H01895 and 17K14834) from the Japan Society for the Promotion of Science (JSPS), and by a grant (No. JPMJCR18R2) from the CREST program of the Japan Science and Technology Agency (JST).

**Conflicts of Interest:** The authors declare no conflict of interest.

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
