# Peer review of "A Cellulose Electrolysis Cell with Metal-Free Carbon Electrodes"

_catalysts, doi:10.3390/catal10010106_

Round 1

Reviewer 1 Report

In this manuscript by Li et al. the authors present an approach to production of hydrogen from the electrolysis of cellulose without the use of a metal based electrocatalyst. The authors used mesoporous carbon as the cathode and a partially oxygenated carbon anode at 150 degrees Celsius for the electrolysis of cellulose. The mesoporous carbon catalyzed the hydrogen evolution on a scale comparable to that using a Pt/C cathode. This is an original study that would be of interest to the readers of Catalysts. As such I recommend accepting this manuscript for publication in its current form.

Reviewer 2 Report

Some minor errors need correction (e.g. line 108: >>Barrett-Joyner-Halanda<< should be: >>Barrett-Joyner-Halenda<<)

Reviewer 3 Report

In this manuscript, Yaorong Li and coworkers have reported on a cathode material labeled as MJ10 for HER. The manuscript is overall well-written and presented. MJ10 as the cathode material shows slightly lower onset voltage for cellulose electrocatalysis compared to that of Pt/C or KB cathode. However, I have few serious concerns about how the experiments were carried out and the materials were synthesized as indicated below. 

Line 45: “The present study describes an inexpensive mesoporous carbon that exhibited HER activity comparable to that obtained for a Pt/C cathode by optimizing the pore and electron structures of the carbon.”

Line 94: “To improve cathode pore structure, mesoporous carbon materials (MH, MJ10, 30, and 150) prepared by a MgO-template method [30,31] were used in place of carbon blacks. The porosity of the four samples was adjusted mainly by controlling the amount of the MgO used.”

Both these statements strongly suggest that the materials were either prepared or modified as a part of this current work. However, in the whole manuscript, no procedure on the preparation/modification was described. Indeed, it appears later that the materials were probably commercially purchased:

Line 162:  “Vulcan XC-72R, KB EC-600JDK, AB, and mesoporous carbons (MH, MJ10, MJ30, and MJ150) were purchased from Cabot (GA, USA), Akzo Nobel (NY, USA), Denka (Tokyo, Japan), and Toyo Tanso (Osaka, Japan), respectively.”

These contradictory statements are misleading and are enough to disqualify this manuscript to be considered for publication in Catalysts. I suggest the authors be more transparent and consistent in presenting the results in the future versions of the manuscript and rewrite the manuscript in accordance with those changes.  

Reviewer 4 Report

How does H3PO4 affect to the total H+ content in the reaction media and thus, H2 generation? Have the authors tried other reagents to hydrolize the cellulose?

Which is the long term performance of these electrodes? In order to verify the metal-free electrodes can present a better performance than Pt-C, long term runs should be performed comparing both of them. 

Round 2

Reviewer 3 Report

Thanks for addressing the concerns. The manuscript is suitable for publication.